# Surface Refractive Surgery Outcomes in Israeli Combat Pilots

**Asaf Achiron** [1,2]📷, **Nadav Shemesh** [1,2,*], **Tal Yahalomi** [3]📷, **Dana Barequet** [1,2], **Amit Biran** [2], **Eliya Levinger** [1,2], **Nadav Levinger** [1,4]📷, **Shmuel Levinger** [1] and **Ami Hirsch** [1]

1   Enaim Refractive Surgery Centers, Jerusalem 9438307, Israel
2   Department of Ophthalmology, Tel Aviv Sourasky Medical Center, Sackler School of Medicine, Tel Aviv University, Tel Aviv 69978, Israel
3   Department of Ophthalmology, Samson Assuta Ashdod Hospital, The Faculty of Health Sciences, Ben-Gurion University of the Negev, Beer-Sheva 7747629, Israel
4   Department of Ophthalmology, Hadassah Medical Center, Jerusalem 91120, Israel
*   Correspondence: nadavshemesh91@gmail.com

**Abstract:** Photorefractive keratectomy (PRK) has long been the method of choice for refractive surgery in pilots, and was FDA approved for U.S. Air Force aviators in 2000. We retrospectively reviewed the medical records of 16 male combat pilots (mean age $25.0 \pm 5.5$ years) who had undergone bilateral laser refractive surgery with surface ablation (alcohol-assisted PRK: 81.25%, transepithelial-PRK: 18.75%), and who had a mean baseline spherical equivalent (SE) of $-2.1 \pm 0.7$ D in the right eye, and $-2.0 \pm 0.7$ D in the left. The mean follow-up was $8.4 \pm 6.6$ months. On the last visit, the uncorrected visual acuity (UCVA) had improved from $0.75 \pm 0.33$ logMar to $-0.02 \pm 0.03$ logMar ($p < 0.001$), and from $0.72 \pm 0.36$ logMar to $-0.02 \pm 0.05$ logMar ($p < 0.001$), for the right and left eyes, respectively. The percentages of participants with a right eye UCVA of at least 0.0, $-0.08$, and $-0.18$ logMAR (6/6, 6/5, and 6/4 Snellen in meters) were 100%, 37.5%, and 6.2%, respectively, and for the left eye, 93.7%, 43.75%, and 6.2%, respectively. No complications occurred. This is the first study to assess refractive surgery outcomes in a cohort of Israeli combat pilots. Surface refractive surgery effectively improved UCVA and reduced spectacle reliance for the members of this visually demanding profession.

**Keywords:** combat; pilots; myopia; refractive surgery

## 1. Introduction

Laser corneal refractive surgery is a widely performed procedure with high patient satisfaction [1–4]. There are two basic approaches: one is to create a corneal flap and then use the excimer laser to perform stromal ablation underneath it (LASIK—Laser-assisted in situ keratomileusis), and the other is to remove the epithelium and apply laser energy directly on the Bowman's membrane (surface ablation). With the second approach, the epithelium can either be removed manually (PRK—Photorefractive keratectomy) or with the excimer laser system (transepithelial-PRK) [5]. More than 200 million procedures have been conducted globally, with hugely increased success rates, thanks to major technological advancements in corneal reshaping excimer laser systems and eye-tracking software [2–4,6].

The transepithelial-PRK method, using an excimer laser to vaporize the corneal tissue, was first introduced in 1985. In 1995, the PRK procedure for refractive surgery was authorized by the Food and Drug Administration (FDA) [7]. Further enhancing the usefulness of the method is that certain drugs, such as l-cysteine, have been shown to enhance corneal healing and reduce corneal haze following the operation [8–11]. PRK is a good option for pilots, soldiers, and individuals with a higher risk of flap dislocation [8], and was approved by the U.S. Air Force (USAF) for aviators in August, 2000 [9]. The LASIK flap's stability in extreme situations (like the very high G-force experienced when in combat aircraft) was in question until just over 10 years ago, when the first cases of pilots who continued to fly trouble-free after LASIK emerged. The topic was studied in depth by both the US Air

Force and the Naval Aerospace Medical Institute of the United States, and approved in 2007 (US Air Force) and 2011 (US Naval Aviators) as a waivable procedure for all classes of aviators. Since then, excellent results have been reported in more than 300 Navy aircraft pilots [10–12]. Even during extreme situations such as ejection from an aircraft, the LASIK flap has been shown to have good stability in an experimental simulation with rabbits [13], and in at least one human [14]. In the German Air Force, personnel who have undergone PRK are allowed to fly as long as they meet the force's visual standards, while post-LASIK personnel are prohibited from flying due to the possible instability of the thinner cornea in a combat aircraft environment [15]. Israeli Air Force personnel are allowed to undergo any refractive procedure, as long as they are over 21 years old and meet the refractive requirements [16].

Numerous large-scale, long-term studies have reviewed the safety and efficacy of refractive surgery in active-duty military personnel (Table 1) [17–22]. Additionally, Stanely et al. have published a review of the experience gained by the US Navy with refractive surgery in general, and particularly in aviators [9]. Only three focus studies have evaluated visual outcomes in combat pilots after PRK (Table 2) [23–25]. Moon et al. noted a lack of data on long-term visual and refractive outcomes after PRK in combat pilots [23].

To remedy the lack of information on the topic, this study aims to examine our long-term experience with refractive surgery in a cohort of Israeli combat pilots.

**Table 1.** Previous studies conducted on post-refractive surgery military personnel.

| Author | Region | No. of Eyes | Follow-Up (Months) | Subjects | Pre-Operative SE (D) | Follow-Up | Post-Operative SE (D) | % of Eyes Achieved ≥20/20 | % of Eyes within ±0.50 D | Comments |
|---|---|---|---|---|---|---|---|---|---|---|
| Tanzer et al. [17] | United States | 651 | 3 | US Naval aviators | −2.56 ± na, 1.86 ± na, −0.34 ± na (myopia, hyperopia, and mixed astigmatism, respectively) | 3 months | na | 98.1%, 100%, and 92.3% (myopia, hyperopia, and mixed astigmatism, respectively) | na | Refractive stability was achieved at 1 month post-surgery. |
| Hammond et al. [18] | United States | 32,068 | na | Soldiers whose mission involves at the line of battle or behind hostile lines. | Na | Na | na | 85.6% | na | - |
| Godiwalla et al. [19] | United States | 160 | 48–204 | Military servicemen. | Na | 4–17 years | na | 99% | 81% | - |
| Schallhorn et al. [20] | United States | 30 | 12 | Active duty Navy/Marine personnel. | −3.35 ± na | 1 year | 0.32 ± 0.53 | 100% | 70% | - |
| Sia et al. [21] | United States | 720 (360 patients) | na | US military service members | −2.97 ± 1.86 | na | Na | 99.7% | Na | - |
| Ang et al. [22] | Singapore | 309 | 12 | Singapore Armed Forces servicemen | −3.33 ± 1.15 | 1 year | −0.03 ± 0.15 | 95.5% | 99.7% | - |

SE; Spherical Equivalent, PRK; photorefractive keratectomy CDVA; corrected distance visual acuity, D; Diopters, UDVA; uncorrected distance visual acuity, LASIK; laser in situ keratomileusis, na; Not available data.

**Table 2.** Previous studies conducted on post-PRK combat pilots.

| Author | Region | No. of Eyes | Follow-Up (Months) | Subjects | Pre-Operative SE (D) | Follow-Up | Post-Operative SE (D) | % of Eyes Achieved ≥20/20 | % of Eyes within ±0.50 D | Comments |
|---|---|---|---|---|---|---|---|---|---|---|
| Moon et al. [26] | Korea | 38 | 48 | Air Force pilots | 1.51 ± 1.15 | 4 years | −0.29 ± 0.51 | 89.5% | 71.1% | The refraction stabilized by 6 months and was maintained up to the 4-year follow-up. |
| See et al. [27] | Singapore | 149 | 12 | Air Force pilot | −3.39 ± 1.19 | 1 year | 0 ± 0.02 | 98.5% | 100% | The cumulative incidence of retreatments was 6.7%. Refractive stability was achieved at 3 months post-surgery. |
| Van de Pol. [28] | United States | 18 | 6 | Black Hawk helicopter pilots | −1.52 ± na | 6 months | na | na | na | Mean UDVA post-operatively was −0.13 ± 0.1 |

SE; Spherical Equivalent, PRK; photorefractive keratectomy CDVA; corrected distance visual acuity, D; Diopters, UDVA; uncorrected distance visual acuity, LASIK; laser in situ keratomileusis, na; Not available data.

## 2. Materials and Methods

### 2.1. Methods

All data for the study were collected and processed according to the norms and procedures of the Tel Aviv Medical Center's Institutional Review Board and the principles described in the Helsinki Declaration, protocol number 0689 17-TLV.

### 2.2. Study Participants

This case-series study included consecutive Israeli Air Force combat pilots who underwent either PRK or transepithelial-PRK, with the same surgeon (AH), at the Enaim Refractive Surgery Center, Tel-Aviv, Israel. Data was obtained from the computerized database registry.

### 2.3. Data Collection

The following demographic and pre-operative information was extracted from the medical files of all eligible pilots: age, gender, refractive error (sphere, cylinder, and spherical equivalents (SE)), keratometry values, pre-operative pachymetry, and pupil size. The following intraoperative information was extracted: eye involved (left or right), treatment zone, ablation depth, and procedural complications. Post-operative information included: refractive error, keratometry values, uncorrected distance visual acuity (UDVA), and corrected distance visual acuity (CDVA). Efficacy was calculated as the ratio of pre-operative CDVA to post-operative UDVA. Safety was calculated as the ratio of pre-operative CDVA to post-operative CDVA. Pre and post-operative pupil sizes and biomechanical parameters were collected using the Peramis aberrometer (CSO, Florence, Italy) and Ocular Response Analyzer (ORA; Reichert Ophthalmic Instruments, Buffalo, NY), respectively.

### 2.4. Surgical Technique

Before surgery, one drop of topical anesthetic (benoxinate hydrochloride 0.4%) was instilled in the conjunctival fornix of the eye. A lid speculum was then inserted. Epithelial removal was performed, either with alcohol (20% ethyl alcohol placed on the cornea for 15 s), or by transepithelial PRK using the Schwind Amaris 1050 (Permis; SCHWIND eye-tech-solutions, Kleinostheim, Germany) for laser ablation. Following excimer ablation, a sponge soaked in 0.02% mitomycin C was placed on the stroma for 20 to 30 s (depending on the amount of ablation). Before placing the contact lens, the mitomycin C solution was rinsed out. Loteprednol was prescribed three times a day for the first month, and two times a day for the second month. The patient was examined one day, one week, and one, three, and six months after surgery.

### 2.5. Statistical Analysis

Statistical analysis was conducted using IBM SPSS software version 23 (Armonk, NY, USA) and MedCalc software version 12.5 (Mariakerke, Belgium). The Shapiro–Wilk test was used to determine whether clinical parameter distributions were normal. We used the Wilcoxon signed-rank test to compare related and unrelated variables since none of the continuous variables were normally distributed. For categorical variables, we used Fisher's exact test. The percentages of eyes with UCVA of at least 0.0, −0.08, and −0.18 logMAR (6/6, 6/5, and 6/4 Snellen in meters) were also calculated. *p*-values of less than 0.05 were considered to be statistically significant in a two-sided test.

## 3. Results

The study included 16 male pilots (mean age $25.0 \pm 5.5$ years) who had undergone bilateral surface refractive surgery (alcohol-assisted PRK: 68.7%, transepithelial-PRK: 31.3%) to correct myopic errors, with a mean baseline SE of $-2.1 \pm 0.7$ D in the right eye and $-2.0 \pm 0.7$ D in the left eye. Baseline demographics and pre- and intraoperative parameters are summarized in Table 3.

**Table 3.** Baseline demographics, pre- and intraoperative parameters.

| | Right Eye | Left Eye |
|---|---|---|
| Corneal hysteresis, mmHg | 9.92 ± 1.48 | 9.89 ± 1.51 |
| Corneal resistance factor, mmHg | 9.22 ± 1.66 | 9.48 ± 1.77 |
| Waveform score | 6.50 ± 1.54 | 5.36 ± 1.75 |
| Pupil size, mm | 6.46 ± 0.66 | 6.50 ± 0.83 |
| K1, D | 43.18 ± 1.65 | 43.24 ± 1.72 |
| K2, D | 44.11 ± 1.84 | 44.25 ± 1.94 |
| Kmean, D | 43.64 ± 1.72 | 43.73 ± 1.79 |
| K Cylinder, D | −0.79 ± 0.75 | −0.71 ± 1.02 |
| Pachymetry at the thinnest point, μm | 525.3 ± 40.9 | 526.9 ± 40.6 |
| Treatment sphere, D | −1.93 ± 0.97 | −1.91 ± 0.96 |
| Treatment cylinder, D | −0.87 ± 0.59 | −0.94 ± 0.66 |
| Target spherical equivalent, D | 0.27 ± 0.17 | 0.28 ± 0.23 |
| Optical zone, mm | 7.01 ± 0.42 | 7.04 ± 0.41 |
| Treatment zone, mm | 1.06 ± 0.27 | 1.05 ± 0.21 |
| Ablation zone, mm | 8.21 ± 0.29 | 8.23 ± 0.26 |
| Central ablation, μm | 60.0 ± 23.3 | 60.4 ± 24.4 |

Patients had a mean follow-up time of 8.4 ± 6.6 months, ranging from 1.02 to 23.74. At the last visit, the UCVA had improved from 0.75 ± 0.33 logMar to −0.02 ± 0.03 logMar, $p < 0.001$ for the right eye and from 0.72 ± 0.36 logMar to −0.02 ± 0.05 logMar, $p < 0.001$ for the left eye. Moreover, sphere and cylinder were examined in the last follow-up visit for 11 out of 16 patients. Mean SE for right and left eyes were 0.23 ± 0.23 and 0.20 ± 0.23, respectively, and Defocus Equivalent (DE) for right and left eyes were 0.27 ± 0.21 and 0.30 ± 0.26, respectively. The percentage of participants having UCVA of at least 0.0, −0.08, and −0.18 logMAR (6/6, 6/5, and 6/4 Snellen in meters) for each eye is shown in Figure 1. The safety index was 1.0 for both eyes, and the efficacy index was 1.0 for the right eye and 0.98 for the left (one eye reached 6/8). Complications, such as corneal haze and persistent epithelial defect, did not arise in any eye, and all subjects resumed their routine flights and training.

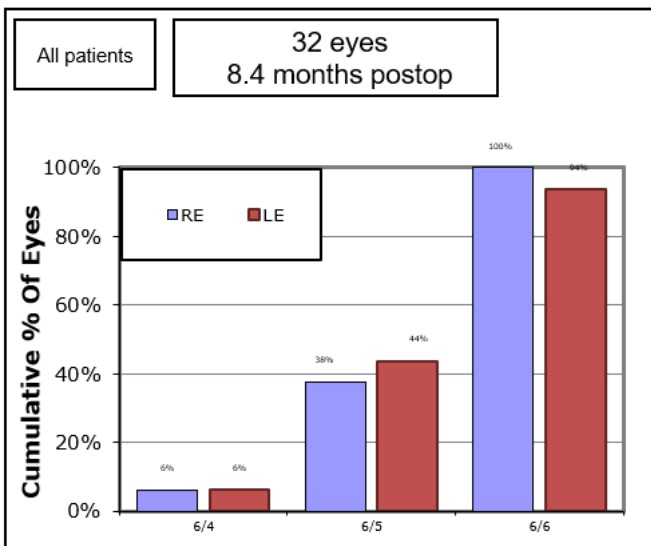

**Figure 1.** Percent of participants having a UCVA of at least 6/6, 6/5, and 6/4.

Comparison between subgroups PRK (N = 13) and transepithelial-PRK (N = 3) of the refractive results showed that postoperative UDVA of 6/6 or better was obtained in 96% and 100% of the eyes, respectively ($p = 0.24$) (Figure 2).

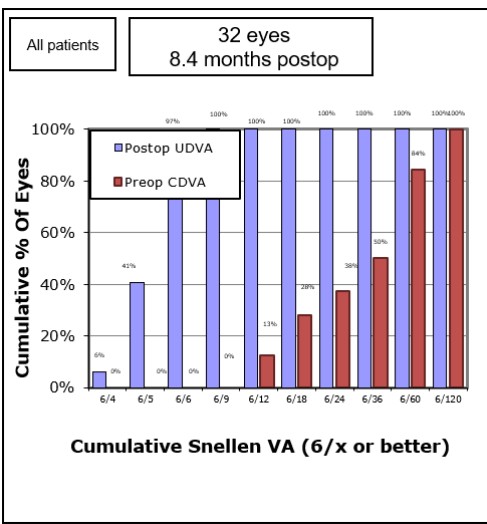

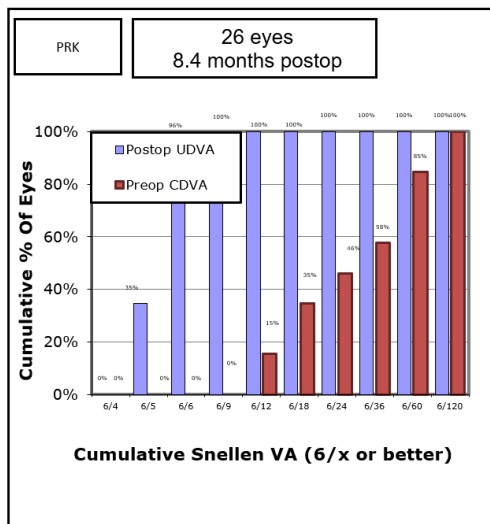

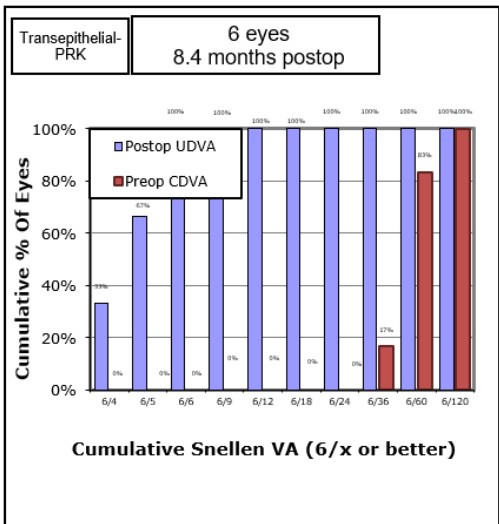

**Figure 2.** Uncorrected distance visual acuity.

## 4. Discussion

Military air crew members must perform under extremely demanding conditions, in a low-visibility setting. The unique environment—including high altitude, low atmospheric

pressure, low oxygen tension, low humidity, high UV light exposure, and high gravity-force load—makes excellent central and peripheral vision essential. Since more than 25% of the general population has refractive errors, which may exclude otherwise optimal candidates from flight careers, refractive surgery is becoming an attractive option for military applicants, expanding the pool of potential aviation trainees. Since its inception in the 2000 s, the Warfighter Refractive Eye Surgery Program has treated over 200,000 U.S. active duty service members using laser surgery [23,24]. The effectiveness and safety of refractive procedures have also been shown in US Navy aviators [13]. Refractive surgery also eliminates the interface issues caused by spectacle frame edges, lens reflections, and glare [28]. High-velocity and low-altitude flights are guided primarily by direct visual input: the jet's sophisticated instrumentation, sensors, warning devices, and automation systems require the pilot to make accurate, time-limited visual discriminations under degraded conditions [18].

Our findings have supported previous reports extolling PRK surgery in air combat pilots. Van de pol et al. prospectively studied 20 Black Hawk helicopter United States Army pilots [28]. They showed a considerable recovery of all visual performance outcomes (day/night, with/without night vision goggles). In addition, 19 of the 20 pilots returned to flight status one month after surgery, with overall flight performance remaining steady or improving from baseline, demonstrating performance resilience. Moon et al. [23] evaluated a four-year, post-PRK follow-up period in 20 Korean pilots with low to moderate myopia. Following their four-year follow-ups, almost 90% of the patients had an uncorrected Snellen vision of 6/6 or better, and 71.1% had emmetropia within 0.50 D. The authors concluded that high-altitude environmental stress exposure does not affect refractive stability after PRK. By six months, the refraction had stabilized, and remained stable until the four-year follow-up stage.

See et al. presented a retrospective case series of 149 eyes of 76 consecutive Singapore Air Force pilots with low to moderate myopia, all of Asian origin [27]. A 12-month follow-up showed that 98.5% of eyes had a UDVA of 0.00 LogMAR, 100.0% of eyes had an SE refraction within 0.50 D of intended correction, and only 2.3% of eyes had a loss of corrected distance visual acuity CDVA of the 0.20 LogMAR. There was a cumulative incidence of 6.7% of retreatments, and a cumulative incidence of 6.0% of grade II or worse corneal haze requiring retreatment.

Kaluzny et al. and Gaeckle's studies compared transepithelial-PRK to PRK and showed that refractive outcomes are similar for the two procedures [25,29]. In our study, a comparison between the two subgroups showed similar refractive results as well, implying that, in terms of refractive outcomes, the two methods are non-inferior to each other.

We acknowledge several limitations to the current study. First, the study's retrospective format makes it inherently prone to selection bias. This is especially true because all patients in our research were subjected to a thorough medical selection procedure and pre-operative ophthalmic screening before having PRK, and because there was no control group. Second, this study contains a limited number of young participants with low to moderate myopia. Third, no contrast sensibility measurements, which may influence the patients' professional performances, were made. Fourth, we present an average follow-up of $8.4 \pm 6.6$ months, and, according to prior research, myopic regression happens most frequently within the first year after surgery [30]. However, a recent 10-year follow-up research study on PRK in low to moderate myopes found that myopic regression in eyes that did not require retreatment was modest during the period from 3 months to 10 years post-surgically, with the mean myopic regression in SE refraction being just 0.49 D [31].

In combat aviation, visual performance is vital to the mission and safety of the aircrew. Only with near-perfect visual acuity can the aviator maintain a visual scan of the inside and outside of the aircraft, day and night, under hypoxic and hypobaric conditions, and when other sensory inputs fail. PRK is a safe, effective, and well-tolerated procedure for correcting refractive errors and reducing spectacle dependence in active Air Force pilots. Its effectiveness has been well documented in civil airmen, military aviators, and

astronauts [32–34]. Our study observed no complications during the follow-up period, and all pilots in this cohort resumed their active flight duties. This information may help open the door to people who, because of their naturally imperfect vision, would not otherwise have the opportunity to train as combat pilots.

**Author Contributions:** Conceptualization, A.A.; methodology, A.A. and A.H.; software, A.A.; validation, S.L. and A.H.; formal analysis, A.A.; investigation, N.S., T.Y., D.B., A.B. and N.L.; writing—original draft preparation, A.A. and N.S.; writing—review and editing, E.L. and S.L.; supervision, E.L., N.L., S.L. and A.H.; project administration, N.S. All authors have read and agreed to the published version of the manuscript.

**Funding:** This research received no external funding.

**Institutional Review Board Statement:** The study was conducted in accordance with the Declaration of Helsinki, and approved by the Institutional Review Board of Tel Aviv Medical Center (protocol code 0689 17-TL, June 2017).

**Informed Consent Statement:** Informed consent was obtained from all subjects involved in the study.

**Data Availability Statement:** Not applicable.

**Conflicts of Interest:** The authors declare no conflict of interest.

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
