# Peer review of "Surface Refractive Surgery Outcomes in Israeli Combat Pilots"

_biomedinformatics, doi:10.3390/biomedinformatics2040046_

Round 1

Reviewer 1 Report

This is an interested work. Congratulations! 

I have missed contrast sensibility measurements.

On the other hand  I consider if you will compare your results with civil people ones, it could be more complet.

So, you could include it in the work limitations if you do not have these dates.

Author Response

  1. Extensive editing of English language and style required – The manuscript was checked and revised by a native English speaker colleague (Danni Meyerson, [email protected]).
  2. I have missed contrast sensibility measurements – No contrast sensibility measurements were conducted. Therefore, we included it in the work limitations “Thirdly, no contrast sensibility measurments, which may have influence of the patients professional performances, were made.”
  3. I consider if you will compare your results with civil people ones, it could be more complete. So, you could include it in the work limitations if you do not have these dates. – No information was extracted regarding civil people. We included it in the work limitations. “This is especially true because all patients in our research were subjected to a thorough medical selection procedure and pre-operative ophthalmic screening before having PRK, and the lack of a control group.”

Reviewer 2 Report

the manuscript is very interesting, it deals with a very delicate and continuously evolving bureaucratic issue. I recommend some changes to make it even more publishable. I recommend extending the introductory part by specifying the other methods for correcting refractive defects. it would be very interesting to know what regulations exist in other states on laser correction in soldiers. I also recommend, if the authors agree, to add some bibliographic notes both in the text and in the chapter of the bibliography. it is a treatment to speed up the closure of the post prk epithelium and limit the risk of haze. I recommend inserting notes number 4 on line 39   Meduri A, Grenga PL, Scorolli L, Ceruti P, Ferreri G. Role of cysteine in corneal wound healing after photorefractive keratectomy. Ophthalmic Res. 2009;41(2):76-82. doi: 10.1159/000187623. Epub 2008 Dec 20. PMID: 19122468.
  Scorolli L, Meduri A, Morara M, Scalinci SZ, Greco P, Meduri RA, Colombati S. Effect of cysteine in transgenic mice on healing of corneal epithelium after excimer laser photoablation. Ophthalmologica. 2008;222(6):380-5. doi: 10.1159/000151691. Epub 2008 Aug 28. PMID: 18753800.
    Meduri A, Bergandi L, Perroni P, Silvagno F, Aragona P. Oral l-Cysteine Supplementation Enhances the Long Term-Effect of Topical Basic Fibroblast Growth Factor (bFGF) in Reducing the Corneal Haze after Photorefractive Keratectomy in Myopic Patients. Pharmaceuticals (Basel). 2020 Apr 15;13(4):67. doi: 10.3390/ph13040067. PMID: 32326563; PMCID: PMC7243117.
  Meduri A, Scorolli L, Scalinci SZ, Grenga PL, Lupo S, Rechichi M, Meduri E. Effect of the combination of basic fibroblast growth factor and cysteine on corneal epithelial healing after photorefractive keratectomy in patients affected by myopia. Indian J Ophthalmol. 2014 Apr;62(4):424-8. doi: 10.4103/0301-4738.119420. PMID: 24145571; PMCID: PMC4064216.
  I also recommend simplifying the chapter on materials and methods.
I recommend giving more emphasis to the conclusions. I recommend improving the English language with a native speaker

Author Response

  1. I recommend some changes to make it even more publishable. I recommend extending the introductory part by specifying the other methods for correcting refractive defects. – Information was added according to the reviewer’s recommendation to the introduction. “There are two basic approaches: first by creating a corneal flap to proceed with stromal ablation with the excimer laser below it (LASIK - Laser-assisted in situ keratomileusis), or removing the epithelium and applying laser energy directly on Bowman's membrane (surface ablation). In the latter alternative, in turn, the epithelium can be removed manually (PRK - Photorefractive keratectomy) or with the same excimer laser system (transepithelial-PRK)”.
  2. It would be very interesting to know what regulations exist in other states on laser correction in soldiers. – Information is stated regarding USA and Germany Air Forces. Morever, information was added regarding Israeli Air Force Israeli Air Force personnel are allowed to undergo any refractive procedure as long as they are over 21 years old and meet the refractive requirements.”
  3.  I recommend inserting notes number 4 on line 39 Meduri A, Grenga PL, Scorolli L, Ceruti P, Ferreri G. Role of cysteine in corneal wound healing after photorefractive keratectomy. Ophthalmic Res. 2009;41(2):76-82. doi: 10.1159/000187623. Epub 2008 Dec 20. PMID: 19122468.
     Scorolli L, Meduri A, Morara M, Scalinci SZ, Greco P, Meduri RA, Colombati S. Effect of cysteine in transgenic mice on healing of corneal epithelium after excimer laser photoablation. Ophthalmologica. 2008;222(6):380-5. doi: 10.1159/000151691. Epub 2008 Aug 28. PMID: 18753800.
     Meduri A, Bergandi L, Perroni P, Silvagno F, Aragona P. Oral l-Cysteine Supplementation Enhances the Long Term-Effect of Topical Basic Fibroblast Growth Factor (bFGF) in Reducing the Corneal Haze after Photorefractive Keratectomy in Myopic Patients. Pharmaceuticals (Basel). 2020 Apr 15;13(4):67. doi: 10.3390/ph13040067. PMID: 32326563; PMCID: PMC7243117.
     Meduri A, Scorolli L, Scalinci SZ, Grenga PL, Lupo S, Rechichi M, Meduri E. Effect of the combination of basic fibroblast growth factor and cysteine on corneal epithelial healing after photorefractive keratectomy in patients affected by myopia. Indian J Ophthalmol. 2014 Apr;62(4):424-8. doi: 10.4103/0301-4738.119420. PMID: 24145571; PMCID: PMC4064216. - The text was modified according to the reviewer’s suggestion, and the references were added. “Moreover, medical treatments, such as l-Cysteine, have been shown to enhance corneal healing and reduce corneal haze following PRK.”
  4.  I also recommend simplifying the chapter on materials and methods. – The text was modified according to the reviewer’s suggestion. Major changes: (1) Study participents part shortened. (2) Surgical technique part was modified. “ Following excimer ablation, a sponge soaked in 0.02% mitomycin C was placed on the stroma for 20 to 30 seconds (depending on the amount of ablation). To place the contact lens, the mitomycin C solution was rinsed. Lotepredinol was prescribed three times a day for the first month, and two times a day for the second month. A patient was examined one day, one week, one, three, and six months following surgery.”
  5. I recommend giving more emphasis to the conclusions. - The conclusion part was modified according to the reviewer’s recommendation"“To conclude, in combat aviation, visual performance is vital to the mission and safety of the aircrew. Near-perfect visual acuity must effectively and efficiently allow the aviator to maintain a visual scan inside and outside the aircraft, day and night, under hypoxic and hypobaric conditions and when other sensory inputs fail. PRK is a safe, effective, and well-tolerated procedure for correcting refractive errors and reducing spectacle dependence in active air force men. It has been well validated in civil airmen, military aviators, and astronauts during space flights [16, 26-27]. Our study observed no complications during the follow-up period, and all pilots in this cohort resumed their active duties of flights. This information may assist decision-makers that are responsible for combat pilots’ regulations, in their decisions for the future. These regulations can open the door for service as combat pilots, for people who did not have this opportunity before.”
  6. I recommend improving the English language with a native speaker. - The manuscript was checked and revised by a native English speaker colleague (Danni Meyerson, [email protected]).

Reviewer 3 Report

Surface refractive surgery outcomes in Israeli combat pilots.

Line 16. It reads: “Using a retrospective chart review of combat pilots who underwent laser refractive surgery in we examined the records of 16 male pilots (mean age 25.0±5.5 years) who underwent bilateral surface refractive surgery (alcohol-assisted PRK: 68.7%, transepithelial-PRK: 31.3%).

COMMENT

Consider modifying to: “We retrospectively reviewed the medical records of 16 male combat pilots (mean age 25.0±5.5 years), who underwent bilateral laser refractive surgery with surface ablation (alcohol-assisted PRK: 68.7%, transepithelial-PRK: 31.3%).”

Line 22. It reads: “The percentage of participants having UCVA of at least 6/6, 6/5, and 6/4 was 100%, 37.5%, and 6.2%, respectively, for the right eye; and 93.7%, 43.75%, and 6.2%, respectively, for the left”.

COMMENT

Consider modifying to: “The percentage of participants having UCVA of at least 0.0, -0.08 and  -0.18 logMAR (6/6, 6/5, and 6/4 Snellen in meters) was 100%, 37.5%, and 6.2%, respectively, for the right eye; and 93.7%, 43.75%, and 6.2%, respectively, for the left”.

In addition, it would be useful for the reader to include in the “Abstract” the mean preoperative spherical equivalent.

Line 32. It reads: “Refractive surgery is a widely performed procedure with high patient satisfaction [1]. With more than 200 million procedures conducted globally, technological advancements in corneal reshaping excimer laser systems and eye-tracking software have resulted in unparalleled success [2].”

COMMENT

The introduction to the subject of keratorefractive surgery is too short, and the difference between LASIK and surface ablation is not described at all. Cited references are not enough.

Consider modifying to: "Laser corneal refractive surgery is a widely performed procedure with high patient satisfaction [1] (Zhang et al, 2022; Hamam et al, 2020; Ang et al, 2021). There are two basic approaches: first creating a corneal flap to proceed with stromal ablation with the excimer laser below it (LASIK), or removing the epithelium and applying laser energy directly on Bowman's membrane (surface ablation). In the latter alternative, in turn, the epithelium can be removed manually (PRK) or with the same excimer laser system (transepithelial-PRK) (Hamam et al, 2020; Rodriguez et al, 2020). With more than 200 million procedures conducted globally, technological advancements in corneal reshaping excimer laser systems and eye-tracking software have resulted in unparalleled success [2] (Zhang et al, 2022; Hamam et al, 2020; Ang et al, 2021).”

Additional references to be cited:

Zhang H, Li M, Cen Z. Excimer Laser Corneal Refractive Surgery in the Clinic: A Systematic Review and Meta-analysis. Comput Math Methods Med. 2022 Jun 15;2022:7130422. doi: 10.1155/2022/7130422. PMID: 35756422; PMCID: PMC9217613.

Hamam KM, Gbreel MI, Elsheikh R, Benmelouka AY, Ouerdane Y, Hassan AK, Hamdallah A, Elsnhory AB, Nourelden AZ, Masoud AT, Ali AA, Ragab KM, Ibrahim AM. Outcome comparison between wavefront-guided and wavefront-optimized photorefractive keratectomy: A systematic review and meta-analysis. Indian J Ophthalmol. 2020 Dec;68(12):2691-2698. doi: 10.4103/ijo.IJO_2921_20. PMID: 33229644; PMCID: PMC7856933.

Ang M, Gatinel D, Reinstein DZ, Mertens E, Alió Del Barrio JL, Alió JL. Refractive surgery beyond 2020. Eye (Lond). 2021 Feb;35(2):362-382. doi: 10.1038/s41433-020-1096-5. Epub 2020 Jul 24. PMID: 32709958; PMCID: PMC8027012.

Rodriguez AH, Galvis V, Tello A, Parra MM, Rojas MÁ, Arba MS, Camacho AP. Fellow eye comparison between alcohol-assisted and single-step transepithelial photorefractive keratectomy: late mid-term outcomes. Rom J Ophthalmol. 2020 Apr-Jun;64(2):176-183. PMID: 32685784; PMCID: PMC7339690.

Line 39. It reads: “In the US, PRK was approved by the U.S. Air Force (USAF) for aviators in August 2000. It has become the most commonly used refractive method for authorized Naval aviators, flying officers, and flight training students [5]. After PRK, military air force personnel in Germany can fly if they meet the appropriate visual standards. Laser-assisted in situ keratomileusis (LASIK) is not allowed due to the thinner cornea's possible instability in the combat aircraft environment after the flap creation [6].”

COMMENT

These statements are not currently correct. Since 2011, when the Naval Aerospace Medical Institute authorized waivers for flight duty following successful LASIK, it has become increasingly popular among naval aviators.

In addition, the sentence “After PRK, military air force personnel in Germany can fly if they meet the appropriate visual standards” is superfluous.

Therefore, consider modifying to.

“In the United States, PRK was approved by the U.S. Air Force (USAF) for aviators in August 2000 [5]. Although initially, until a little over 10 years ago, the stability of the LASIK flap in the extreme situations of very high level of G-forces imposed by the flight conditions of a combat aircraft, was in question, after some cases of pilots who continued to carry out their activities without problems after LASIK, the topic was studied in detail by the Naval Aerospace Medical Institute from the United States, and since 2011, this institute approved LASIK as a waiverable procedure for Naval aviators, and excellent results have been reported in more than 300 Navy fighter aircraft pilots (Levy et al 2003; Tanzer et al. 2013). Even in extreme situations, like during an aircraft ejection, good stability of the LASIK flap has been shown experimentally in rabbits (Goodman et al 2003), and at least in one human (Richmond et al 2016). However, there is at least a theoretical a higher risk than a very high G-force may have effect in optical aberrations in a cornea biomechanically more weakened by the flap cut, and therefore surface ablations are preferred by some armed forces from several countries in combat pilots (Rauscher et al, 2008).”

References to be cited:

Levy Y, Zadok D, Barenboim E. Laser in situ keratomileusis in a combat jet aircraft pilot. J Cataract Refract Surg. 2003 Jun;29(6):1239-41. doi: 10.1016/s0886-3350(02)01995-8. PMID: 12842699.

Tanzer DJ, Brunstetter T, Zeber R, Hofmeister E, Kaupp S, Kelly N, Mirzaoff M, Sray W, Brown M, Schallhorn S. Laser in situ keratomileusis in United States Naval aviators. J Cataract Refract Surg 2013; 39:1047–1058.

Goodman RL, Johnson DA, Dillon H, Edelhauser HF, Waller SG. Laser in situ keratomileusis flap stability during simulated aircraft ejection in a rabbit model. Cornea 2003; 22:142–145

Richmond CJ, Barker PD, Levine EM, Hofmeister EM. Laser in situ keratomileusis flap stability in an aviator following aircraft ejection. J Cataract Refract Surg. 2016 Nov;42(11):1681-1683. doi: 10.1016/j.jcrs.2016.10.001. PMID: 27956297.

Line 46. It reads: “While numerous, large, long-term studies reviewed the safety and efficacy of refractive surgery in active-duty military personnel [5, 7-11], combat pilots were the group of focus in only three studies evaluating visual outcomes after PRK (Table 1) [12-14].

COMMENT:

In their review published in 2008, Stanley et al. described the results of a study in Naval pilots: Schallhorn SC, Tidwell, JL, Brown M, et al. Photorefractive keratectomy in

Naval aviation. Paper presented at the Aerospace Medical Association Annual

Meeting; 15 May 2006; Orlando, Florida, USA.

Furthermore, Tanzer et al published their results of LASIK in pilots in 2013.

Stanley PF, Tanzer DJ, Schallhorn SC. Laser refractive surgery in the United States Navy. Curr Opin Ophthalmol. 2008 Jul;19(4):321-4. doi: 10.1097/ICU.0b013e3283009ee3. PMID: 18545015.

Tanzer DJ, Brunstetter T, Zeber R, Hofmeister E, Kaupp S, Kelly N, Mirzaoff M, Sray W, Brown M, Schallhorn S. Laser in situ keratomileusis in United States Naval aviators. J Cataract Refract Surg 2013; 39:1047–1058.

Line 73. It reads: “(sphere, spherical equivalence (SE), and cylinder), keratometry values, pre-operative pachymetry, and pupil size.”.

COMMENT

It should read: “(sphere, cylinder, and spherical equivalent [SE])”.

It is necessary to indicate what devices were used to determine keratometry values, pre-operative pachymetry, and pupil size.

Line 80. It reads: “Pupil size was collected from the Schwind Amaris 1050 (Permis; SCHWIND eye-tech-solutions, Kleinostheim, Germany)”

COMMENT

The Schwind Amaris 1050 does not report pupillometry. It seems to be a typo, and that the pupillometry information was taken from the Peramis aberrometer.

 Therefore, it should read: ““Pupil size was collected from the Peramis aberrometer (CSO, Florence, Italy)”

Line 93. It reads: “Patients were prescribed moxifloxacin 0.5% (six times a day), dexamethasone 0.1% (two or four times a day), and artificial tears (four times a day).”

COMMENT

Did some patient receive dexamethasone 0.1% only twice a day? What were the criteria to indicate it 2 or 4 times a day?

Moreover, it is necessary to clarify for how long do the patients received dexamethasone 0.1% and if it was or nor tapered.

Line 108. It reads: “16 male pilots (mean age 25.0±5.5 years) who underwent bilateral surface refractive surgery (alcohol-assisted PRK:68.7%, transepithelial-PRK: 31.3%) to correct a mean baseline SE of -2.1±0.7D on the right eye and -2.0±0.7D on the left eye”.

COMMENT

It should read: “16 male pilots (mean age 25.0±5.5 years) who underwent bilateral surface refractive surgery (alcohol-assisted PRK:68.7%, transepithelial-PRK: 31.3%) to correct myopic errors, with a mean baseline SE of -2.1±0.7D on the right eye and -2.0±0.7D on the left eye”.

Table 2.

It reads: “Target spherical equivalent, D 0.27±0.17 0.28±0.23”.

COMMENT

It seems to be a typo, since the target spherical equivalent in these patients should be “Plano”, i.e. 0.0 D. Or were they planned for having a residual refractive error? Or did the authors mean that they used a nomogram?

Line 116. It reads: “Patients had a mean follow-up time of 8.4±6.6 months. At the last visit, the UCVA had improved from 0.75±0.33 logMar to -0.02±0.03 logMar, p<0.001 for the right eye and from 0.72±0.36 logMar to -0.02±0.05 logMar, p<0.001 for the left eye”.

COMMENT

What was the range of follow-up time?

What was the sphere and the cylinder at the last follow-up visit? It would be useful to calculate also both Spherical equivalent and defocus equivalent.

Line 119. It reads: “The percentage of participants having UCVA of at least 6/6, 6/5, and 6/4 was 100%, 37.5%, and 6.2%, respectively, for the right eye; and 93.7%, 43.75%, and 6.2%, respectively, for the left”.

COMMENT

Since the values are shown in Figure 2, consider modifying to: “The percentage of participants having UCVA of at least 0.0, -0.08 and -0.18 logMAR (6/6, 6/5, and 6/4 Snellen in meters) for each eye, are shown in Figure 2”.

Figure 1 does not provide important additional information to what is indicated in the text, so it is actually superfluous and should be deleted.

Line 131, it reads: “This is the first study to assess refractive surgery outcomes in a cohort of Israeli combat pilots. PRK was effective in improving UCVA and reducing spectacle dependence in this professional group with high visual demands”.

COMMENT

It should read: “This is the first study to assess refractive surgery outcomes in a cohort of Israeli combat pilots. PRK  and transepithelial-PRK were effective in improving UCVA and reducing spectacle dependence in this professional group with high visual demands”.

ADDITIONAL COMMENTS:

It is necessary to comment in “Discussion” section about the study by Tanzer et al about LASIK in combat pilots (Tanzer DJ, Brunstetter T, Zeber R, Hofmeister E, Kaupp S, Kelly N, Mirzaoff M, Sray W, Brown M, Schallhorn S. Laser in situ keratomileusis in United States Naval aviators. J Cataract Refract Surg 2013; 39:1047–1058).

It is necessary to comment in “Discussion” section about published studies on comparative results between PRK and transepithelial-PRK, and to compare the results in the two subgroups of the cohort in the present study.

Reference:

Rodriguez AH, Galvis V, Tello A, Parra MM, Rojas MÁ, Arba MS, Camacho AP. Fellow eye comparison between alcohol-assisted and single-step transepithelial photorefractive keratectomy: late mid-term outcomes. Rom J Ophthalmol. 2020 Apr-Jun;64(2):176-183. PMID: 32685784; PMCID: PMC7339690

Author Response

  1. Line 16. Consider modifying to: “We retrospectively reviewed the medical records of 16 male combat pilots (mean age 25.0±5.5 years), who underwent bilateral laser refractive surgery with surface ablation (alcohol-assisted PRK: 68.7%, transepithelial-PRK: 31.3%).” – The text was modified according to the reviewer’s suggestion. “We retrospectively reviewed the medical records of 16 male combat pilots (mean age 25.0±5.5 years), who underwent bilateral laser refractive surgery with surface ablation (alcohol-assisted PRK: 68.7%, transepithelial-PRK: 31.3%), with mean baseline spherical Equivalent (SE) of -2.1±0.7D on the right eye and -2.0±0.7D on the left eye.”
  2. Line 22. Consider modifying to: “The percentage of participants having UCVA of at least 0.0, -0.08 and -0.18 logMAR (6/6, 6/5, and 6/4 Snellen in meters) was 100%, 37.5%, and 6.2%, respectively, for the right eye; and 93.7%, 43.75%, and 6.2%, respectively, for the left”. - The text was modified according to the reviewer’s recommendation, both in the abstract and the results. “The percentage of participants having UCVA of at least 0.0, -0.08 and -0.18 logMAR (6/6, 6/5, and 6/4 Snellen in meters) was 100%, 37.5%, and 6.2%, respectively, for the right eye; and 93.7%, 43.75%, and 6.2%, respectively, for the left.”
  3. It would be useful for the reader to include in the “Abstract” the mean preoperative spherical equivalent – The information was added in the abstract: “with mean baseline Spherical Equivalent (SE) of -2.1±0.7D on the right eye and -2.0±0.7D on the left eye.”
  4. Line 32. Consider modifying to: "Laser corneal refractive surgery is a widely performed procedure with high patient satisfaction [1] (Zhang et al, 2022; Hamam et al, 2020; Ang et al, 2021). There are two basic approaches: first creating a corneal flap to proceed with stromal ablation with the excimer laser below it (LASIK), or removing the epithelium and applying laser energy directly on Bowman's membrane (surface ablation). In the latter alternative, in turn, the epithelium can be removed manually (PRK) or with the same excimer laser system (transepithelial-PRK) (Hamam et al, 2020; Rodriguez et al, 2020). With more than 200 million procedures conducted globally, technological advancements in corneal reshaping excimer laser systems and eye-tracking software have resulted in unparalleled success [2] (Zhang et al, 2022; Hamam et al, 2020; Ang et al, 2021).” – The text and references were modified according to the reviewer recommendations.:“Laser corneal refractive surgery is a widely performed procedure with high patient satisfaction. There are two basic approaches: first by creating a corneal flap to proceed with stromal ablation with the excimer laser below it (LASIK - Laser-assisted in situ keratomileusis), or removing the epithelium and applying laser energy directly on Bowman's membrane (surface ablation). In the latter alternative, in turn, the epithelium can be removed manually (PRK - Photorefractive keratectomy) or with the same excimer laser system (transepithelial-PRK). With more than 200 million procedures conducted globally, technological advancements in corneal reshaping excimer laser systems and eye-tracking software have increased success rates to unprecedented levels”
  5. Line 39. These statements are not currently correct. Since 2011, when the Naval Aerospace Medical Institute authorized waivers for flight duty following successful LASIK, it has become increasingly popular among naval aviators – The text and references were modified according to the reviewer’s recommendations:In the United States, PRK was approved by the U.S. Air Force (USAF) for aviators in August 2000. Although initially, until a little over 10 years ago, the stability of the LASIK flap in extreme situations of very high levels of G-forces imposed by the flight conditions of a combat aircraft, was in question, after some cases of pilots who continued to carry out their activities without problems after LASIK. The topic was studied in detail by the US Air Force and Naval Aerospace Medical Institute from the United States, and since 2007 (US airforce) and 2011 (US Naval Aviators), LASIK is approved as a waiverable procedure for all classes of aviators, and excellent results have been reported in more than 300 Navy aircraft pilots. Even during extreme situations, like an aircraft ejection, good stability of the LASIK flap has been shown experimentally in rabbits, and at least in one human. In the German air force, personnel who underwent PRK are allowed to fly if they meet the appropriate visual standards, while post-LASIK personnel is prohibited due to the thinner cornea’s possible instability in a combat aircraft environment. Israeli Air Force personnel are allowed to undergo any refractive procedure as long as they are over 21 years old and meet the refractive requirements.”
  6. Line 39. The sentence “After PRK, military air force personnel in Germany can fly if they meet the appropriate visual standards” is superfluous. – The text and references were modified, in order to make them more relevant to the text.: “In the German air force, personnel who underwent PRK are allowed to fly if they meet the appropriate visual standards, while post-LASIK personnel are prohibited due to the thinner cornea’s possible instability in a combat aircraft environment.”
  7. Line 46. In their review published in 2008, Stanley et al. described the results of a study in Naval pilots: Schallhorn SC, Tidwell, JL, Brown M, et al. Photorefractive keratectomy in Naval aviation. Paper presented at the Aerospace Medical Association Annual. Meeting; 15 May 2006; Orlando, Florida, USA. Furthermore, Tanzer et al published their results of LASIK in pilots in 2013. – The text and references were modified according to the reviewer’s recommendations. Alongside this, a table concluding studies regarding refractive procedures in military personnel was added.
  8. Line 73. It should read: “(sphere, cylinder, and spherical equivalent [SE])”. – The text and references were modified according to the reviewer’s recommendations. “ (sphere, cylinder, and spherical equivalence (SE))”
  9. Line 73. It is necessary to indicate what devices were used to determine keratometry values, pre-operative pachymetry, and pupil size. – The information is stated at the end of the paragraph. “Pre and post-operative pupil size and biomechanical parameters were collected from the Schwind Amaris 1050 (Permis; SCHWIND eye-tech-solutions, Kleinostheim, Germany) and Ocular Response Analyzer (ORA; Reichert Ophthalmic Instruments, Buffalo, NY), respectively.“
  10. Line 80. The Schwind Amaris 1050 does not report pupillometry. It seems to be a typo, and that the pupillometry information was taken from the Peramis aberrometer. Therefore, it should read: ““Pupil size was collected from the Peramis aberrometer (CSO, Florence, Italy)”- The text was modified according to the reviewer’s recommendation. Pre and post-operative pupil size and biomechanical parameters were collected from the Peramis aberrometer (CSO, Florence, Italy) and Ocular Response Analyzer (ORA; Reichert Ophthalmic Instruments, Buffalo, NY), respectively.
  11. Line 93. Did some patient receive dexamethasone 0.1% only twice a day? What were the criteria to indicate it 2 or 4 times a day? Moreover, it is necessary to clarify for how long do the patients received dexamethasone 0.1% and if it was or nor tapered. - The text was modified according to the reviewer’s recommendation. “Lotepredinol was prescribed three times a day for the first month, and two times a day for the second month”.
  12. Line 108. It should read: “16 male pilots (mean age 25.0±5.5 years) who underwent bilateral surface refractive surgery (alcohol-assisted PRK:68.7%, transepithelial-PRK: 31.3%) to correct myopic errors, with a mean baseline SE of -2.1±0.7D on the right eye and -2.0±0.7D on the left eye”. - The text and references were modified according to the reviewer’s recommendations: “16 male pilots (mean age 25.0±5.5 years) who underwent bilateral surface refractive surgery (alcohol-assisted PRK:68.7%, transepithelial-PRK: 31.3%) to correct myopic errors, with a mean baseline SE of -2.1±0.7D on the right eye and -2.0±0.7D on the left eye”
  13. Table 2. It seems to be a typo, since the target spherical equivalent in these patients should be “Plano”, i.e. 0.0 D. Or were they planned for having a residual refractive error? Or did the authors mean that they used a nomogram? –Based on our age-related normogram we usually target fo +0.4- 0.2.
  14. Line 116. What was the range of follow-up time? The text was modified according to the reviewer’s recommendation:“Patients had a mean follow-up time of 8.4±6.6 months, ranges from 1.02 to 23.74.”
  15. Line 116. What was the sphere and the cylinder at the last follow-up visit? It would be useful to calculate also both Spherical equivalent and defocus equivalent. The text was modified according to the reviewer’s recommendation. “Moreover, sphere and cylinder were examined in the last follow-up visit for 11 out of 16 patients. Mean SE for right and left eyes were 0.23±0.23 and 0.20±0.23, respectively, and Defocus Equivalent (DE) for right and left eyes were 0.27±0.21 and 0.30±0.26, respectively.”
  16. Line 119. Since the values are shown in Figure 2, consider modifying to: “The percentage of participants having UCVA of at least 0.0, -0.08 and -0.18 logMAR (6/6, 6/5, and 6/4 Snellen in meters) for each eye, are shown in Figure 2”. - The text and references were modified according to the reviewer’s recommendations. The percentage of participants having UCVA of at least 0.0, -0.08, and -0.18 logMAR (6/6, 6/5, and 6/4 Snellen in meters) for each eye as shown in Figure 1 (Due to the deletion of Figure 1).
  17. Figure 1 does not provide important additional information to what is indicated in the text, so it is actually superfluous and should be deleted. – The figure was deleted.
  18. Line 131. It should read: “This is the first study to assess refractive surgery outcomes in a cohort of Israeli combat pilots. PRK and transepithelial-PRK were effective in improving UCVA and reducing spectacle dependence in this professional group with high visual demands”. - The text and references were modified according to the reviewer’s recommendations.:“This is the first study to assess refractive surgery outcomes in a cohort of Israeli combat pilots. PRK and transepithelial PRK were effective in improving UCVA and reducing spectacle dependence in this professional group with high visual demands.”
  19.  It is necessary to comment in “Discussion” section about the study by Tanzer et al about LASIK in combat pilots (Tanzer DJ, Brunstetter T, Zeber R, Hofmeister E, Kaupp S, Kelly N, Mirzaoff M, Sray W, Brown M, Schallhorn S. Laser in situ keratomileusis in United States Naval aviators. J Cataract Refract Surg 2013; 39:1047–1058). – The information was added to the discussion according to the reviewer’s recommendation: “The effectivness and safety of refractive procedures were also shown on naval US naval aviators.”
  20. It is necessary to comment in the “Discussion” section about published studies on comparative results between PRK and transepithelial-PRK, and to compare the results in the two subgroups of the cohort in the present study. The text and references were modified according to the reviewer’s recommendations. Results: “Subgroup comparison, between PRK (N=13) and transepithelial-PRK (N=3) of refractive result showed that postoperative UDVA of 6/6 or better was obtained in 96% and 100% of the eyes respectively (p=0.24) (Figure 2).” Discussion: “Kaluzny et al. and Gaeckle compared in their studies transepithelial-PRK to PRK and showed that refractive outcomes are similar between the two procedures. In our study, a comparison between the two subgroups showed similar refractive results as well, implying the two methods are non-inferior compared to each other, regarding refractive outcomes.”

Round 2

Reviewer 3 Report

Line 5, following the subtitle “Statistical Analysis”. It reads: “The percentages of eyes with 6/6, 6/5, and 6/4 post-operatively were also calculated.”

COMMENT

It should read: “The percentages of eyes with UCVA of at least 0.0, -0.08, and -0.18 logMAR (6/6, 6/5, and 6/4  Snellen in meters) were also calculated.”

Author Response

  1. Following the subtitle “Statistical Analysis, It should read: “The percentages of eyes with UCVA of at least 0.0, -0.08, and -0.18 logMAR (6/6, 6/5, and 6/4 Snellen in meters) were also calculated.” - The text was modified according to the reviewer’s recommendation.